# Microplastics in Cetaceans Stranded on the Portuguese Coast

**DOI:** 10.3390/ani13203263

**Published:** 2023-10-19

**Authors:** Sara Sá, Andreia Torres-Pereira, Marisa Ferreira, Sílvia S. Monteiro, Raquel Fradoca, Marina Sequeira, José Vingada, Catarina Eira

**Affiliations:** 1Department of Biology & CESAM & ECOMARE/CPRAM, Universidade de Aveiro, 3810-193 Aveiro, Portugal; aftpereira@ua.pt (A.T.-P.); s.monteiro@ua.pt (S.S.M.); catarina.eira@ua.pt (C.E.); 2Portuguese Wildlife Society (SPVS), Estação de Campo de Quiaios, 3081-101 Figueira da Foz, Portugal; mctferreira@socpvs.org (M.F.); rfradoca@ua.pt (R.F.); spvs@socpvs.org (J.V.); 3Instituto da Conservação da Natureza e Florestas (ICNF), Av. da República 16, 1050-191 Lisboa, Portugal; marina.sequeira@icnf.pt

**Keywords:** marine mammals, marine litter, ingestion, contaminants, northeast Atlantic

## Abstract

**Simple Summary:**

This study characterises microplastics ingested by small cetaceans on the coast of Portugal. The intestine contents of 38 stranded cetaceans were processed in the laboratory to remove as much organic matter as possible from the samples and facilitate the detection of microplastics under a stereomicroscope. This study evaluated the possible influence of several biological and health variables (e.g., species, sex, body condition) on the amount of microplastics found in three small cetacean species, particularly on common dolphins, due to the larger number of available samples. Most of the analysed individuals had microplastics, with harbour porpoises revealing a significantly higher median number of microplastics than common dolphins, probably due to their different diets, use of habitat and feeding strategies. However, none of the other tested variables significantly influenced the number of microplastics in either all of the analysed species or in the common dolphin group. The relatively low numbers of microplastics found in the present study should not be enough to cause physical and chemical sublethal effects, although the potential effects of plastic-derived pollutants are not yet completely understood. Future monitoring of biota should rely on improved and standardised protocols for microplastic analyses.

**Abstract:**

This study characterises microplastics in small cetaceans on the coast of Portugal and assesses the relationship between several biological variables and the amount of detected microplastics. The intestines of 38 stranded dead cetaceans were processed in the laboratory, with digestion methods adapted to the amount of organic matter in each sample. The influence of several biological and health variables (e.g., species, sex, body condition) on the amount of microplastics was tested in all analysed species and particularly in common dolphins, due to the larger number of available samples. Most of the analysed individuals had microplastics in the intestine (92.11%), with harbour porpoises revealing a significantly higher median number of microplastics than common dolphins, probably due to their different diets, use of habitat and feeding strategies. None of the other tested variables significantly influenced the number of microplastics. Moreover, the microplastics found should not be enough to cause physical or chemical sublethal effects, although the correlation between microplastic ingestion and plastic additive bioaccumulation in cetacean tissues requires further investigation. Future monitoring in biota should rely on improved and standardised protocols for microplastic analyses in complex samples to allow for accurate analyses of larger samples and spatio-temporal comparisons.

## 1. Introduction

Due to the high societal demand for plastic objects and its high persistence [1,2,3], plastic litter is accumulating throughout the marine environment, affecting marine fauna mainly through entanglement and ingestion [4,5,6]. Within plastic litter, microplastics (i.e., smaller than 5 mm [7]), which can be either intentionally produced in small sizes (e.g., pellets, microbeads) or result from the fragmentation of larger plastic items during long periods of time [8,9] are receiving increasing attention and concern [10,11]. Mean abundance levels of microplastics in the North Atlantic region were only exceeded by mean abundances reported for the Mediterranean Sea and North Pacific [12]. Microplastics’ small size makes them bioavailable to a wide range of marine organisms, from zooplankton [13,14] to top predators [15,16,17,18,19,20]. Additionally, the toxic chemical additives in microplastics and/or the contaminants (e.g., persistent organic pollutants, heavy metals) that they adsorb from the surrounding seawater may leach into the animal tissues and bioaccumulate along the trophic chain [21,22,23], potentially causing harmful effects [24,25]. Several studies, mostly on marine invertebrates and fish, report direct physical harm (e.g., intestinal damage or blockage [26,27]), but also toxic effects such as growth inhibition [28,29], decreased feeding and energy reserves [30,31], decreased reproductive output [32,33], oxidative stress and immunological alterations [28,34,35], neuro- and liver toxicity [36,37], and endocrine disruption [38,39]. As most of the reported effects of microplastic ingestion result from studies performed under controlled laboratory conditions with model species from low trophic levels (e.g., crustaceans, molluscs and fishes), little is known about the consequences in the field for marine top predators [16,17,22].

Marine mammals, due to their high trophic level, long lifespan, and propensity to accumulate environmental contaminants, are considered important sentinels of marine ecosystem health, and more recently they have been used to monitor marine plastic pollution [40,41,42]. Considering the existing threats affecting cetacean populations (e.g., by-catch, anthropogenic noise, hunting) [43], it is important to understand whether microplastic pollution poses an additional risk to these populations [41]. The assessment of additional risks is particularly needed on the Atlantic Iberian coast, given the important cetacean bycatch mortality in the area, especially in the case of the critically endangered Iberian porpoise (*Phocoena phocoena*) population [44,45]. To obtain results on plastic litter affecting cetaceans, it has been recommended that national marine mammal stranding networks play an active role in the collection of samples for marine litter analyses and increase the studies on the impacts of marine plastic pollution on cetaceans [41,46,47].

Publication of data on interaction rates and mortality of cetacean species associated with marine litter is useful and necessary to monitor the application of international agreements and European Commission directives (e.g., MSFD 2008/56/EC). However, few studies so far have addressed marine microplastics in cetaceans, probably due to methodological challenges such as the lack of a standardised protocol to analyse microplastic ingestion in the digestive tract of stranded organisms and the large volume of gastric contents to analyse [48,49]. Only a few studies have been developed in the north-east Atlantic [19,48,50,51], and in many areas, no assessment on microplastic ingestion by cetaceans has ever been done. This study characterises microplastics in the intestinal contents of small cetaceans stranded on the coast of Portugal and assesses the relationship between several biological and health-related variables and the amount of microplastics in the analysed individuals.

## 2. Materials and Methods

### 2.1. Study Area

The Portuguese continental coast is 860 km long, ranging from Caminha (41°50′ N, 8°50′ W) to Vila Real St. António (37°12′ N, 7°25′ W). The western coast diverges from the southern coast mainly due to their different topographic and oceanographic characteristics [52]. The north-central western coast is a 310 km section between Caminha and Peniche (Cape Carvoeiro) that presents a wider and flat continental shelf (40–70 km) and a strong, homogeneous upwelling with a peak from July to September [53], thus differentiating this coastal area from the rest of the Portuguese coast. This upwelling is characterised by northern wind regimes, resulting in colder waters with high productivity [53]. Primary consumers tend to aggregate in high-productivity areas, ultimately leading to high abundances of marine predators, including cetaceans [54,55]. In fact, the north-central western coast includes a Site of Community Importance (SCI, NATURA 2000) Maceda—Praia da Vieira (PTCON0063), a marine area dedicated to the protection of cetaceans, particularly harbour porpoises and bottlenose dolphins [56].

### 2.2. Sample Collection

In this study, we analysed 38 intestinal contents from fresh and moderately decomposed cetaceans [57,58] stranded dead on the north-central Portuguese coast. Intestinal contents from harbour porpoises (n = 8), common dolphins (*Delphinus delphis*) (n = 24) and striped dolphins (*Stenella coeroleoalba*) (n = 6) were sampled between 2016 and 2019. The stranded cetaceans were collected by licensed technicians of the regional marine animal stranding network (coordinated nationally by the Institute of Nature Conservation and Forests, ICNF). During post-mortem analyses following a standardised protocol [58], information on species, sex, total body length (TBL), decomposition state, stranding date and location were recorded. Other recorded parameters included age class (calf, juvenile and adult) and maturity state (mature vs. immature) [59,60], body condition (good, moderate, thin and skeletal) [58,61], presence of parasites, and cause of death (bycatch and others, including disease and trauma) [58,62,63] (see Table 1). For those animals with no data on maturity, total body length was used as a proxy for age and maturity [59,60]. For striped dolphin males, the common dolphin maturity value was used, since maturity analyses were not performed.

During the necropsies, the intestines were removed, carefully opened, emptied into glass jars using filtered tap water, immediately sealed (to minimise aerial contamination) and frozen (−20 °C) at the Marine Animal Tissue Bank (license code 13PT0124/S).

All stranding locations were plotted in QGIS 3.10.12-A Coruña using the WGS 1984 data and then projected using the UTM zone 29N projection.

### 2.3. Sample Processing

In the laboratory, after sample thawing, an organic matter digestion method was adapted to the amount of organic matter in each sample, since it may confound the detection of microplastics [64,65]. In samples with low organic matter concentrations, a simple vacuum filtering into a VWR^®^ G693 glass filter (1.2 µm pore; 47 mm diameter) was performed. In samples presenting high organic matter contents, a two-step digestion procedure was adapted from [66]. The digestion solutions included 10% (*w*/*v*) KOH and 15% (*v*/*v*) H_2_O_2_ (35%), and both were prepared and diluted in distilled water. This two-step digestion approach was applied either in the filter containing the sample (if the organic matter content allowed an initial filtration step—protocol A) or applied directly on the sample (if the organic matter content prevented initial sample filtration—protocol B). In protocol A, after sample filtration, 20 mL of 10% KOH was added to the filter with the sample (placed on a Petri dish) and incubated at 50 °C for 1 h (adapted from [64]). Afterwards, the content of the filter was washed, transferred to a new filter, placed on a Petri dish and exposed to 20 mL of 15% H_2_O_2_ at 50 °C for 1 h (adapted from [64]). For protocol B, the supernatant (floating phase) of the samples was filtered onto a VWR^®^ G693 glass filter (1.2 µm; 47 mm diameter) and the remaining sample (solid phase) was transferred to a clean glass beaker and placed in a drying oven at 40 °C until dried. Afterwards, a 10% KOH solution at a ratio of 1:3 (sample: KOH volume) [67,68] was added to the dried solid-phase samples and placed in the oven for 24 h at 60 °C, a temperature previously used in other studies [69,70], for higher digestion efficiencies. Then, the floating phase was vacuum-filtered and the remaining solid phase was placed in an oven at 40 °C until dried. Subsequently, a solution of 15% H_2_O_2_ at a ratio of 1:3 (sample: H_2_O_2_ volume) was added to the remaining dried solid phase and placed in an oven at 50 °C for 1 h (conditions as in [64]), followed by a final filtration step. Each filter was placed in a Petri dish and placed to dry in an oven at 40 °C.

The sample filters were examined under a stereomicroscope (Optika SZM-LED2, Optika Microscopes, Ponteranica, BG, Italy), and particles exhibiting the appearance of microplastics (as described in [71]) were photographed using the eyepiece image analysis system with DinoCapture 2.0 software and subjected to the hot needle test [72,73,74,75] to be validated as plastic material. The particles were categorised according to their type, shape and colour [76] and their largest cross-section (size) was measured. The lowest limit of resolution and detection of microplastics was 0.159 mm.

### 2.4. Quality Assurance and Quality Control (QA/QC) Procedures

The laboratory work (sampling, extraction and identification) was carried out by only one person using a cotton lab coat and blue nitrile gloves on a clean bench in a closed and restricted-access laboratory. Moreover, the bench surface was previously wiped down with 70% ethanol. All materials used during sampling, extraction and processing (including the preparation of digestion solutions) were washed with distilled water before being used and covered while not in use to prevent contamination. During sample processing, all flasks were covered to prevent airborne contamination. However, as we cannot rule out airborne contamination, procedural air blanks were carried out at the sampling, extraction and identification stages. Clean glass microfibre filters in Petri dishes were placed on the laboratory bench while opening and rinsing the intestines during sample processing (e.g., digestion procedures) and also during observation and identification under the stereomicroscope. Microplastics found in procedural air blanks were also examined under the stereomicroscope, measured and submitted to the hot needle test. The number of microplastics found in each air blank was subtracted from the total number of microplastics of the respective sample when characteristics (type, colour, size) were similar.

### 2.5. Data Analysis

The relative frequency (%N) of each microplastic category for the analysed cetaceans was calculated as the number of items of each microplastic category in relation to the total number of microplastics found. The frequency of occurrence (F.O.%) corresponds to the percentage of analysed individuals that contained microplastics in relation to all individuals (F.O.%).

Data on the number of microplastics per individual were non-normally distributed and therefore non-parametric tests were used. The explanatory variables used in this study were: sex (male/female), maturity state (mature/immature), presence/absence of parasites, causes of death (by-catch vs. trauma/disease), sampling years (2017–2019), age classes (calves, juveniles and adults), body condition (good, moderate and thin), and species (common dolphin, harbour porpoise and striped dolphin). The influence of the explanatory variables was tested using the Mann–Whitney U test [77] and Kruskal–Wallis test [78]. Regarding age classes, only common dolphins and harbour porpoises were considered, since we had no data for striped dolphins. Since a larger sample was available for common dolphins (n = 24), the influence of the explanatory variables mentioned above (excluding species) on the number of microplastics and number of fibres found in this species was also tested using the Mann–Whitney U test [77] or the Kruskal–Wallis test [78]. If significant differences among groups were detected in the Kruskal–Wallis test (*p*-value < 0.05), a Dunn’s test [79] was performed. The correlations between individuals’ total body lengths and the number of microplastics in the intestinal contents were assessed using the Spearman rank correlation test. The same test was used to assess any existing correlation between sampling years and the number of microplastics. All tests were performed using the ‘stats’ and ‘rstatix’ packages in the statistical software R v. 4.2.2 [80].

## 3. Results

No macroplastics were found in the intestines of any of the analysed individuals; only meso- (size: 5 ≤ 25 mm, [76]) and microplastics (size: ≤5 mm, [76]) were found. Marine litter particles, including micro- and mesoparticles, were found in 37 of the 38 analysed cetaceans (97.37%). The mean number of particles per animal was 7.05 (±6.45 SD) ranging between 0 and 27 particles per animal. In total, 268 plastic particles were found and 254 of these were microplastics. The mean size of plastics found in our study was 2.01 mm (±1.73 SD), with a range of 0.159 to 13.069 mm. Regarding mesoplastics, the items found were mainly fibres (n = 10; size range: 5.031–13.069 mm), as well as three films (size range: 5.629–7.939 mm) and one fragment (green angular fragment with 9.419 mm). The predominant shape of mesoplastic items was elongated (92.86%) and the most abundant colour was transparent (28.57%), followed by blue, green, white and red with 14.29%. Two common dolphins had mesoplastics only in the intestine.

Considering only microplastics, 92.11% (n = 35) of all individuals had items in the intestines (Table 2), with a mean number of 6.68 (±6.53 SD) (range: 0–27) particles per individual.

### 3.1. Microplastic Types

Details regarding the type of microplastics found in the present study, together with the respective %N and %FO, can be found in Table 2. Microfibres were the most abundant microplastic type, being present in 86.84% (n = 33) of all analysed individuals (Table 2). A total of 195 microfibres (range: 0–27 per individual) were found in the present study (Table 2). With respect to fragments, a total of 44 fragments (range: 0–18 per individual) were found in 31.58% (n = 12) (Table 2) of the analysed individuals. In the case of microplastic films, 14 items were present in 21.05% (n = 8) (Table 2) of the analysed intestinal contents. The number of films per individual varied between 0 and 3 items. The least abundant microplastic category was fibre cluster, with only one white item with 4.797 mm found in the intestine of a common dolphin.

### 3.2. Microplastics According to Cetacean Species

The three analysed cetacean species revealed high frequencies of occurrence of microplastics, with values ranging from 87.50% to 100% (Table 2). The mean number of microplastics per individual was relatively higher in harbour porpoises, followed by striped dolphins (Table 2). In fact, the median number of all microplastic items was significantly different between species (Kruskal–Wallis test, χ^2^ = 8.3487; df = 2; *p*-value = 0.01539), with harbour porpoises showing a significantly higher median number of microplastics (Dunn’s post hoc test, *p*-value: 0.0348) (Figure 1). No significant differences were found in the number of fibres between species, although the *p*-value was marginal (Kruskal–Wallis test, number of fibres: χ^2^ = 5.8395; df = 2; *p*-value = 0.05395).

Microfibres were the dominant type of microplastic in common dolphins and harbour porpoises, representing 84.75% and 80.49% of the total number of items found, respectively (Table 2, Appendix A). The maximum number of 27 fibres was found in a common dolphin. Fragments were the second-most frequent microplastic type in common dolphins (8.47%) and harbour porpoises (13.41%), followed by films (5.93% and 6.10%, respectively). In striped dolphins, although fibres were the most abundant microplastic type (53.70%), fragments were relatively more representative in terms of number (42.59%) than in common dolphins and harbour porpoises (Table 2). Despite the small number of analysed striped dolphins (n = 6), fragments were found in most of them (83.33%, n = 5), whereas they occurred in a lower proportion of common dolphins (20.83%, n = 5) and porpoises (25.00%, n = 2). In fact, 18 fragments were found in one of the analysed striped dolphins (Table 2), a considerably higher number in comparison to the other analysed individuals.

### 3.3. Microplastics According to Biological and Health-Related Features

None of the analysed biological and health-related variables significantly influenced the number of microplastics and fibres found in the overall samples (all individuals) or in common dolphin samples (number of microplastics, *p*-value > 0.05; number of fibres, *p*-value > 0.05) (see Appendix A). Also, no correlations were found between the number of microplastics and individuals’ total length (r_s_ = 0.025; *p*-value = 0.8829) or individuals’ sampling years (r_s_ = 0.033; *p*-value = 0.8418) when considering the overall dataset.

### 3.4. Microplastic Colour

Considering all microplastics recorded in the present study, 13 different colours were identified (Figure 2). Blue was the most frequent colour (27.17%), followed by black (20.47%) and light-coloured items (16.14%) (including transparent, white and beige). Blue was the predominant colour in microplastics in common dolphins (25.42%) and harbour porpoises (29.27%), while in striped dolphins black was the most frequent colour (40.74%). With respect to microfibres, they were mainly blue (34.87%), followed by the red colour (12.31%). Fragments were predominantly black (72.73%) and microplastic films were mostly transparent and grey (21.43% each).

### 3.5. Microplastic Sizes and Shape

With respect to particle size, the majority of microplastics were considered large microplastics (64.17%; 1–≤5 mm; [76]), while small microplastics (0.100–≤ 1 mm; [76]) accounted for 35.83%. Microplastics in the size range of 0.501–1.000 mm were the most abundant, followed by size intervals 1.001–1.500 mm and 2.001–2.500 mm (Appendix A). Elongated was the most common shape among all microplastics found (80,31%).

With respect to microfibres, the mean size was 1.879 mm (±1.121 SD), varying from 0.159 to 4.915 mm. All microfibres had an elongated shape. With respect to fragments, a mean size of 0.981 mm (±0.799 SD) was obtained with a size range from 0.257 to 4.528 mm. Fragments were predominantly irregularly shaped (54.55%). Film sizes varied from 0.622 to 4.741 mm with a mean size of 1.757 mm (±1.1905 SD), and 50% were irregularly shaped.

## 4. Discussion

This is the first study assessing microplastic abundance in the digestive tract of cetaceans stranded on the Portuguese coast and the second one in the Atlantic Iberian Peninsula, since the occurrence of microplastics was reported in stomachs of common dolphins stranded in Galicia (north-western Spain) [51]. Also, our study provides the first results on microplastics in Iberian harbour porpoises.

In this study, 268 particles (micro- and mesoplastics) were found in the intestines of 38 stranded cetaceans, of which 254 were microplastics (≤5 mm). Only a few studies on microplastics in cetaceans are available (see Table 3; [48,68,81]). The number of microplastics found in the present study was similar to the values reported in the UK [48], where 273 particles (including 261 microplastics) were found in the digestive tract (stomachs and intestines) of 50 stranded marine mammals (43 cetaceans; 7 pinnipeds). On the other hand, the values in the present study were lower than the total number of microplastics found in stomach contents of 35 common dolphins stranded in Galicia (n = 411; [51]), in the entire digestive tracts of 21 cetaceans (19 delphinids and two beaked whales) stranded in Ireland (n = 598; [19]), in 12 odontocete individuals stranded in the Macaronesia region (Madeira and Canary Islands) (n = 722; [50]) and in 43 striped dolphins stranded along the Valencian Community, on the Mediterranean coast of Spain (n = 672; [82]) (see Table 3). Differences in sample size, the analysed digestive tract compartments and methodologies may explain this variation and, therefore, comparisons between studies represent a challenging task. In fact, one study on three *Sousa chinensis* individuals [83] analysing only the intestines portion of the digestive tract, as in the present study, reported a total number of microplastics (n = 77) lower than ours. However, care must be taken when comparing results from a considerable different number of sampled individuals. Also, the only other study reporting the total number of microplastics in the intestines reported 38 items in seven belugas from the Beaufort Sea, although values for stomachs and faeces were also reported [81].

Overall, microplastics were found in 35 out of the 38 stranded cetaceans analysed (92.11%) in the present study, which is in accordance with similar studies in the Northeast Atlantic that found microplastics in 100% of the analysed individuals [19,48,50,51] (Table 3). In the western Mediterranean Sea, 90.5% of the analysed striped dolphin digestive tracts (43 individuals) contained microplastics [82] and in the Beaufort Sea, microplastics were detected in the digestive tracts (stomach and intestine) of all the analysed belugas (7 individuals) [81]. The mean number of particles (micro- and mesoparticles) per individual in our study, 7.05 ± 6.45 SD, was higher than that obtained in the intestines of cetaceans stranded in the UK (1.7 ± 1.4 SD) [48] and slightly higher than the number of microplastics found in the intestines of the seven belugas from the Beaufort Sea (5.43 ± 2.64 SD) [81]. However, our mean value appears to be lower than the mean number found in the stomachs of common dolphins stranded in Galicia [51] and in digestive tracts of striped dolphins from the western Mediterranean Sea [82] (Table 3). In particular, the mean number of microplastics detected per individual in the intestines analysed in present study was much lower compared to the numbers found in the digestive tract in cetaceans stranded in Macaronesia (e.g., fibres 59.08 ± 40.52 SD) [50]. However, these higher mean values are not surprising considering that these studies analysed other compartments besides intestines (e.g., stomachs [51,82] that seem to partially retain the microplastics) and usually present higher abundances [17,48].

The most frequent colours among microplastics were blue (27.17%) and black (20.47%), as in other studies [48,51]. In Ireland, blue was also identified as the predominant colour in the digestive tracts (oesophagus, stomachs and intestines) of the analysed cetaceans, however, grey was the second most abundant colour, followed by black [19]. Once again, the study from the Macaronesia region [50] stands out, reporting green as the most abundant colour, followed by red. Black microplastics were predominant in the digestive tracts of striped dolphins analysed on the Spanish eastern coast, as for the microplastics found in the striped dolphin intestines analysed in our study [82].

The most abundant microplastic type in our study was microfibres (representing 76.77%), which is in line with previously reported environmental microplastic concentrations in subtidal sediments [84], in open ocean subsurface water [85] and marine biota intestinal contents [83], including studies on the Portuguese coast [86,87,88]. Particularly, our result is also in line with other studies analysing ingestion of microplastics by cetaceans and other marine top predators, such as seabirds and sea turtles [19,48,50,51,68,82,83,89]. However, in stomachs and intestines of belugas from the Beaufort Sea [81] fragments were the most abundant microplastic type (51%), followed by fibres (49%) (Table 3). In the present study, fragments occurred in five out of the six analysed striped dolphins. In this species, fragments were relatively more representative (42.59%) than in common dolphins and harbour porpoises (Table 2). The mean number of fragments found per striped dolphin in our study (Table 2, 3.83 ± 6.97 SD) was similar to the mean number of fragments per individual (3.00 ± 1.15 SD) in the digestive tract of cetaceans stranded in Macaronesia [50]. This was unexpected since the latter study analysed all compartments of the digestive tract of the sampled cetaceans. The fact that striped dolphins are more associated with offshore areas of Portuguese waters [90] might be related with a higher ingestion of fragments, since plastic litter with high floatability is known to accumulate in the large-scale subtropical offshore convergence zones, where litter may remain for long periods of time while degrading [91,92,93]. Fragments found in this study probably originated from the breakdown of larger plastic pieces, since they were predominantly irregularly shaped and all of them presented more rough than smooth edges [94,95], similarly to microplastics found in Portuguese coastal waters [96]. No pellets or microbeads were found during microplastic identification.

Considering the mean size of plastics, no macroplastics were found in the intestines analysed in the present study, as in the intestines analysed in China [83], while macroplastics were found in the intestines of cetaceans stranded in Ireland [19]. When considering microplastics only (1.728 mm ± 1.140 SD), our value was within the range reported for microplastics in intestinal tracts of *Sousa chinensis*, 2.2 mm ± 0.4 SD [83]. Regarding size classes, microplastics in the size range of 0.501–1.000 mm were the most abundant in the present study, as in marine mammals in the UK [48]. Notice that the most abundant microplastics size class in marine mammals in Ireland was 1.0–2.0 mm [19]. Also, the mean size of all microfibres (≤5 mm) (1.879 mm ± 1.121 SD) as well as the mean size of microfibres in the 24 analysed common dolphins in our study (1.910 mm ± 1.057 SD) were similar to microfibre mean size in common dolphin stomachs in Galicia (2.11 mm ± 1.26 SD) [51]. The fragments mean size (0.98 mm ± 0.80 SD) in our study was within the mean values obtained in the digestive tracts (stomachs and intestines) of cetaceans stranded in the UK (0.9 mm ± 1.1 SD) [48], and it was slightly lower than the mean size of fragments found in common dolphin stomachs in Galicia (1.29 mm ± 0.93 SD) [51].

Microplastics may be ingested by cetaceans directly from the seawater (primary ingestion), or indirectly through ingestion of contaminated prey items, which is known as trophic transfer (secondary ingestion) [19,48,68]. Therefore, it can be quite difficult to understand which of the exposure routes led to the ingestion of microplastics. It was suggested that the exposure route in marine mammals depends on the species predominant feeding strategy [17]. However, different feeding strategies often interrelate and are employed in conjunction by individual species [97]. For example, although harbour porpoises are predominantly suction feeders, they also separate prey items from the water before swallowing it using their elastic pharynx and external throat grooves, expelling the water back through the mouth [98]. An important point supporting the “secondary ingestion” hypothesis is that microplastics have been detected in several fish species (pelagic and demersal) collected in Portuguese waters [99,100,101,102,103,104,105]. In turn, these fish species have been found in the diet of common dolphins, harbour porpoises and striped dolphins in Portugal [106,107,108,109,110]. More specifically, the ingestion of microplastics by fish in Portuguese waters was evaluated in the European hake (*Merluccius merluccius*; F.O. = 29%), European sardine (*Sardina pilchardus;* F.O. = 0–75%), Atlantic mackerel (*Scomber scombrus*; F.O. = 31%), Atlantic chub mackerel (*Scomber colias;* F.O. = 64–70%), Atlantic horse mackerel (*Trachurus trachurus,* F.O. = 7–100%), European anchovy (*Engraulis encrasicolus*, F.O. = 79–93%) [99,100,101,102,103] and flathead grey mullet (*Mugil cephalus*) (F.O. = 98%; [104]) and fibres were consistently the most abundant microplastic type in all studies. Also, blue was the predominant colour in most fish studies [99,100,101,102]. Therefore, the predominant microplastic type and colour in Portuguese fish commonly predated by cetaceans in Portugal [106,107,108,109,110] correspond to the characteristics of microplastics found in the cetacean species analysed in the present study. Evidence of microplastic transfer across trophic levels was also reported in a study that analysed the presence of microplastics in captive seal scats and in the wild-caught fish they ate [17]. In the present study, results must be interpreted with caution due to the small sample. Nonetheless, the median number of microplastics in harbour porpoises was higher than in common dolphins. These differences are probably associated with differences in diets and feeding strategies of these two species. Studies on small cetaceans’ diets on the Portuguese coast revealed a higher use of demersal fish species by the harbour porpoise, while the common dolphin fed mainly on pelagic and mesopelagic species [106,107,108,109]. Moreover, while common dolphins capture their prey using raptorial-like methods and use suction only to aid in swallowing [99], harbour porpoises are thought to be ‘capture suction feeders’, using suction both to capture the prey and to aid in swallowing [111]. In this way, accidental ingestion of marine litter, including microplastics, when feeding on demersal fish species on or near the seafloor through suction may occur for harbour porpoises [18]. In fact, natural non-food items (e.g., shells, stones, sand) have been found in the stomach contents of harbour porpoises (e.g., [18,112]), confirming that accidental ingestion of items occurs during feeding. These studies also revealed a higher frequency and abundance of natural benthic non-food items in the stomachs of harbour porpoises that had ingested litter, as well as a correlation between the presence of plastic litter and the mass of demersal fishes ingested [18]. There are also studies reporting that demersal fish species ingested a higher number of microplastics than pelagic fish (e.g., [113,114]), which would be in line with the known plastic litter (including microplastics) accumulation in the seafloor [115,116]. However, other studies indicated opposite results or the nonexistence of a relationship between microplastic ingestion by fish and their distribution in the water column [117,118,119].

The presence of microplastics in cetaceans’ intestines suggests that these items may be excreted through faeces along with hard prey structures (fish bones, otoliths and squid beaks), as reported by other studies identifying microplastics in seal scats and in cetacean intestines [17,19,68,120]. However, translocation of microplastics into tissues (blubber, melon, acoustic fat pads and lung) through the intestinal wall was recently detected in marine mammals [121]. The levels of microplastics ingested by the analysed individuals should not be enough to cause physical and chemical sublethal effects, as in similar studies [48,51,82], it is not yet understood to what extent microplastics act as a vector of toxic pollutants (e.g., organochlorides and heavy metals) from the seawater into the tissues of marine mammals and the effects they may have [19,122]. Phthalate (plastic additives) concentrations, used as proxies for plastic litter ingestion, have been recently reported in the blubber of blue and fin whales [123,124,125], in the urine of bottlenose dolphins [126], in the liver of harbour porpoises [127] and in muscle samples of several odontocete species [50]. Nevertheless, the correlation between the presence of microplastics and the bioaccumulation of toxic pollutants in cetacean tissues is still poorly understood, emphasising the need for further research on the mechanisms of metabolisation, accumulation and excretion of pollutants (including microplastics) in cetaceans, particularly in populations with concerning conservation status and declining trends, such as the Iberian harbour porpoise population.

## 5. Conclusions

Microplastic ingestion occurs regularly in the harbour porpoise (*Phocoena phocoena*), common dolphin (*Delphinus delphis*) and striped dolphin (*Stenella coeruleoalba*) in Portuguese waters. Harbour porpoises revealed a significantly higher number of microplastics than common dolphins, probably due to their different diets, use of habitat and feeding strategies. Although most of the analysed individuals had microplastics in the intestine, the relatively low numbers found in the present study should not be enough to cause physical or chemical sublethal effects. Nevertheless, future monitoring of biota from different ocean basins across long time periods should rely on improved and standardised protocols for microplastic analyses in complex samples (see [128]), which allow for accurate analyses of more individuals in order to produce spatio-temporal comparisons. Future research in cetaceans should involve the assessment of physical and chemical properties of the ingested plastic items to determine which are the most problematic. Plastic additive concentrations (phthalates and bisphenols) and other contaminants potentially adsorbed by microplastics should also be assessed to better understand the correlation between microplastic ingestion and pollutant bioaccumulation in these top marine predators.

## Figures and Tables

**Figure 1 animals-13-03263-f001:**
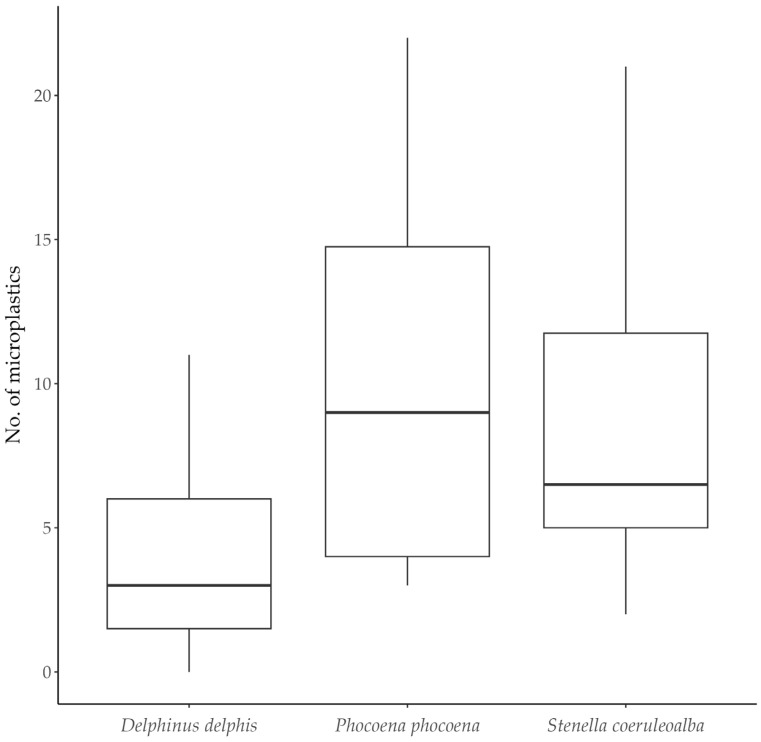
Box plot showing the median number of microplastics detected in each species. The box stretches from the 25th to the 75th percentile (IQR, interquartile range). The line across the box represents the median, and the ends of the vertical line indicate the minimum and maximum values.

**Figure 2 animals-13-03263-f002:**
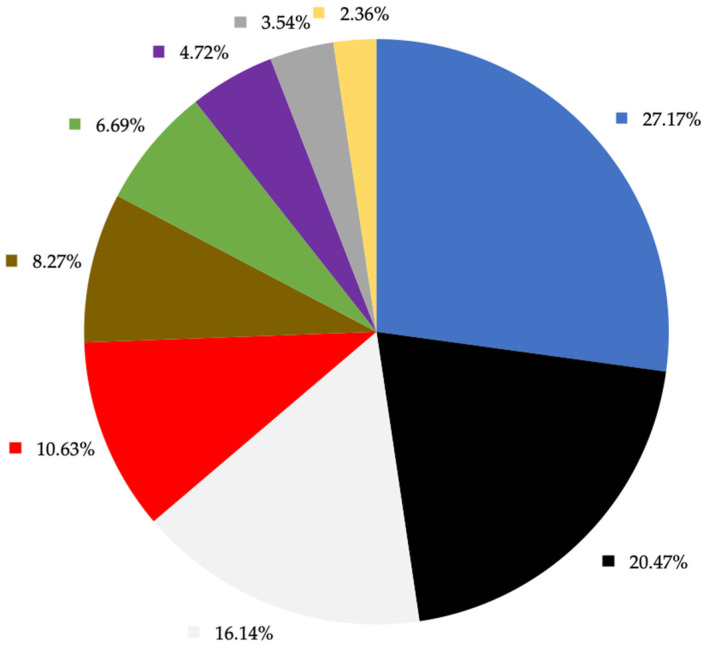
Proportion of microplastic colours found in all analysed cetaceans.

**Table 1 animals-13-03263-t001:** Data collected on each individual analysed in the present study. Species, year of stranding, TBL (total body length, cm), sex, age class, sexual maturity status, body condition, presence of parasites and cause of death. ID, sample number; M, male; F, female: nd, not determined.

ID	Species	Year	TBL (cm)	Sex	Age	Maturity	Body Condition	Parasites	Cause of Death
1	*Delphinus delphis*	2016	−130 *	F	nd	Immature	Moderate	No	Bycatch
2	*Delphinus delphis*	2016	184	M	Juvenile	Immature	Good	Yes	Bycatch
3	*Delphinus delphis*	2017	210	M	Adult	Mature	Moderate	Yes	Bycatch
4	*Delphinus delphis*	2017	192	F	Adult	Mature	Thin	No	Disease
5	*Delphinus delphis*	2017	174.5	M	Juvenile	Immature	Good	Yes	Bycatch
6	*Delphinus delphis*	2017	125	M	Calf	Immature	Moderate	Yes	Bycatch
7	*Delphinus delphis*	2017	132	M	Calf	Immature	Good	No	Bycatch
8	*Delphinus delphis*	2017	130	M	Calf	Immature	Good	Yes	Bycatch
9	*Delphinus delphis*	2017	150	M	Juvenile	Immature	Good	No	Bycatch
10	*Delphinus delphis*	2017	145	M	Juvenile	Immature	Good	Yes	Bycatch
11	*Delphinus delphis*	2017	173	F	Juvenile	Immature	Thin	Yes	Bycatch
12	*Delphinus delphis*	2017	175.5	F	Juvenile	Immature	Moderate	Yes	Disease
13	*Delphinus delphis*	2017	192	F	Adult	Mature	Thin	Yes	Disease
14	*Delphinus delphis*	2017	196	F	Adult	Mature	Thin	Yes	Disease
15	*Delphinus delphis*	2018	−125 *	F	nd	Immature	Good	Yes	Bycatch
16	*Delphinus delphis*	2018	141	F	Juvenile	Immature	Good	No	Trauma
17	*Delphinus delphis*	2018	135	F	Calf	Immature	Good	Yes	Bycatch
18	*Delphinus delphis*	2018	119	M	Calf	Immature	Moderate	No	Bycatch
19	*Delphinus delphis*	2019	138	M	Calf	Immature	Moderate	Yes	Bycatch
20	*Delphinus delphis*	2019	167	M	Juvenile	Immature	Moderate	Yes	Bycatch
21	*Delphinus delphis*	2019	181	F	Juvenile	Immature	Moderate	Yes	Bycatch
22	*Delphinus delphis*	2019	125	F	Calf	Immature	Good	No	Bycatch
23	*Delphinus delphis*	2019	158	M	Juvenile	Immature	Skeletal	Yes	Disease
24	*Delphinus delphis*	2019	196	F	Adult	Mature	nd	nd	Bycatch
25	*Phocoena phocoena*	2017	146	F	Juvenile	Immature	Good	Yes	Bycatch
26	*Phocoena phocoena*	2017	144	F	Juvenile	Immature	Moderate	Yes	Bycatch
27	*Phocoena phocoena*	2017	154	M	Juvenile	Immature	Thin	Yes	Bycatch
28	*Phocoena phocoena*	2017	174	M	Adult	Mature	Thin	Yes	Bycatch
29	*Phocoena phocoena*	2017	147	M	Juvenile	Immature	Thin	Yes	Bycatch
30	*Phocoena phocoena*	2017	150	F	Juvenile	Immature	Moderate	Yes	Bycatch
31	*Phocoena phocoena*	2017	156.5	F	Juvenile	Immature	Good	Yes	Trauma
32	*Phocoena phocoena*	2018	136	F	Juvenile	Immature	Good	Yes	Bycatch
33	*Stenella coeruleoalba*	2017	175	F	nd	Immature	Thin	Yes	Disease
34	*Stenella coeruleoalba*	2018	178	F	nd	Immature	nd	nd	Disease
35	*Stenella coeruleoalba*	2018	176	M	nd	Immature	nd	Yes	Disease
36	*Stenella coeruleoalba*	2018	159	M	nd	Immature	Thin	Yes	Disease
37	*Stenella coeruleoalba*	2018	136	F	nd	Immature	Thin	Yes	Disease
38	*Stenella coeruleoalba*	2019	136	M	nd	Immature	Moderate	No	nd

* Incomplete body length due to caudal extremity removed by instrument.

**Table 2 animals-13-03263-t002:** Number (N), relative frequency (%N) and frequency of occurrence (F.O.%) of each microplastic category for the analysed cetaceans (n = 38). The mean and median numbers of microplastics per individual are also presented. SD, standard deviation. IQR, interquartile range.

Species	Type	N (%N)	F.O.%	Mean (±SD)	Median (IQR)	Range
*D. delphis*	Fibres	100 (84.75)	79.17	4.17 ± 5.75	2.5 (1.0–5.5)	0–27
	Fragments	10 (8.47)	20.83	0.42 ± 1.02	0 (0)	0–4
	Films	7 (5.93)	20.83	0.29 ± 0.62	0 (0)	0–2
	Fibre clusters	1 (0.85)	4.17	-	-	0–1
	Total	118 (100)	87.50	4.92 ± 5.76	3 (1.75–6.25)	0–27
*P. phocoena*	Fibres	66 (80.49)	100	8.25 ± 6.27	5 (3.75–14.50)	2–17
	Fragments	11 (13.41)	25.00	1.38 ± 2.88	0 (0–0.75)	0–8
	Films	5 (6.10)	25.00	0.63 ± 1.19	0 (0–0.50)	0–3
	Fibre clusters	0	0	-	-	-
	Total	82 (100)	100	10.25 ± 7.21	9 (4.0–14.75)	3–22
*S. coeruleoalba*	Fibres	29 (53.70)	100	4.83 ± 4.49	3 (2.25–6.00)	1–13
	Fragments	23 (42.59)	83.33	3.83 ± 6.97	1 (1–1.75)	0–18
	Films	2 (3.70)	16.67	0.33 ± 0.82	0 (0)	0–2
	Fibre clusters	0	0	-	-	-
	Total	54 (100)	100	9.00 ± 6.96	6.5 (5.0–11.75)	2–21
All individuals	Fibres	195 (76.77)	86.84	5.13 ± 5.79	3 (1.0–7.0)	0–27
	Fragments	44 (17.32)	31.58	1.16 ± 3.21	0 (0–1)	0–18
	Films	14 (5.51)	21.05	0.37 ± 0.79	0 (0)	0–3
	Fibre clusters	1 (0.39)	2.63	-	-	-
	Total	254 (100)	92.11	6.68 ± 6.53	4.5 (2.25–9.50)	0–27

**Table 3 animals-13-03263-t003:** Review of published information on meso- and microplastics found in digestive tracts of odontocete species. Documented study area, analysed species, total number of ingested microplastics (or micro- and mesoplastics), frequency of occurrence (F.O.%), mean number of particles found per individual, predominant colour, predominant size class, predominant microplastic type (and respective mean size), digestive tract compartments analysed and references. na, not available.

Study area	Species	Plastic Particles							Reference
		Number	% F.O.	Mean Number ± SD (Range)	Colour	Size Class (mm)	Mean Size ± SD mm	Digestive Compartment	
NE Atlantic (Portugal)									
	common dolphin (24) harbour porpoise (8) striped dolphin (6)	254 micro	92.11	7.05 ± 6.45 micro/mesoplastics (0–27)	blue, black	0.5–1.0	micro/mesoplastics, 2.01 ± 1.73 microfibres, 1.88 ± 1.12 micro/meso, 2.14 ± 1.68	intestine	Present study
NE Atlantic, Macaronesia (Canary and Madeira archipelagos)									
	striped dolphins (5) bottlenose dolphin (2) Risso’s dolphins (2) Short-finned pilot whale (1) Pygmy sperm whale (1) Fraser’s dolphin (1)	722 * micro/meso	100	59.08 ± 40.52 fibres 3.00 ± 1.15 fragments	green, red	na	micro/meso, 2.66 ± 2.51	oesophagus, stomach, duodenal ampulla and intestine	[50]
W Mediterranean (Valencia, Spain)									
	striped dolphin (43)	672 micro	90.5	14.9 ± 22.3 (0–82)	black, red	na	microfibres, no size	digestive tract	[82]
Arctic (Beaufort Sea)									
	belugas (7)	81 micro	100	11.6 ± 6.6	na	na	fragments, no size	stomach intestine	[81]
NE Atlantic (UK)									
	cetaceans (43) pinnipeds (7)	261 micro	100	5.5 ± 2.7 (1–12)	blue, black	0.5–1.0	micro/meso, 2.00 ± 2.30	stomachs and intestines	[48]
W Pacific Ocean, South China Sea (Guangxi Beibu Gulf, China)									
	Indo-Pacific humpback dolphin (3)	77 micro	100	na	white, blue	na	microfibres, 2.20 ± 0.40	intestine	[83]
NE Atlantic (Galicia, Spain)									
	common dolphins (35)	411 micro/meso	100	12.0 ± 8.0 (3–41)	blue, black	na	microfibres, 2.11 ± 1.26	stomachs	[51]
NE Atlantic (Ireland)									
	delphinids (19) beaked whales (2)	598 micro/meso	100	(1–88)	blue, grey	1.0–2.0	microfibres, no size	oesophagus, stomachs, intestine	[19]
NE Atlantic (Ireland)									
	True’s beaked whale (1)	88 micro/meso	-	na	na	na	micro/mesofibres, 2.16 ± 1.39	stomach, intestine	[68]

* out of the 722 particles, 12 were subject to confirmation test.

## Data Availability

The data presented in this study are available on request from the corresponding author. The data are not publicly available because they were obtained under particular data sharing protocols, and they are still in use by the corresponding author.

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
