# Peer review of "Microplastics in Cetaceans Stranded on the Portuguese Coast"

_animals, 2023, doi:10.3390/ani13203263_

Round 1
Reviewer 1 Report
1. The number of references were not following the order. Such as the 18th reference.
2. Compare with the multiple case studies, review papers published in recent years were more suitable for the basic knowledge in the Introduction. Such as:
10.1007/s44169-023-00036-y
10.1016/j.scitotenv.2022.158745
10.1016/j.jhazmat.2022.130274
10.1016/j.envpol.2022.119911
3. Line 63, just focus on the plastic pollution please.
4. Line 81, Portugal is not the novelty of this manuscript. Authors must explain their meaning better.
5. Information mentioned in the Section 2.1 seems not meaningful.
6. Relevant license should be shown in another file.
7. Figure 1 is not informative.
8. There must be some better way to show data in the Table 1.
9. Section 2.5 is good but can be shorten.
10. Table 2, similar with Table 1, improve it. So as Figure 2, 3.
11. The discussion part has a low merit. Maybe it should not separate from the Results. Besides, I can not see some new knowledge from this part, or in the Conclusion.
Reviewer 2 Report
1. This is a nice, thorough report on a good and useful study. The authors include a lot of information and I am satisfied with the methods they used in their analysis. I like all the results they present regarding the sizes, shapes, and colors of microplastics in the intestinal contents.
2. I had many other questions as I was reading the manuscript, and I was pleased that the authors anticipated and addressed my questions. My first questions was about plastic fragments being ingested directly versus through trophic transfer (that is, the plastics being in the diet of the cetaceans). This is very difficult to determine with certainty, and the authors have a nice discussion (in the last three paragraphs of the Discussion section) about feeding methods of the three odontocete species surveyed in their study, and how this might affect ingestion of microplastics. They also cite several studies documenting plastics in prey species of these cetaceans. Still, I wonder if there are additional ways to determine (perhaps from the plastic pieces themselves?) whether they were ingested directly or via prey. Again, I understand that this would be very hard to determine.
3. My next question was whether the small pieces of plastic (meso- and micro-) were primary or secondary fragments—that is, were they in their original form (e.g., nurdles used for injection molding, or tiny microbeads or other pellets in a product) or were they created by the breakdown and erosion of larger pieces of plastic? This could be determined through microscopy (mostly, if not definitively), by looking at the form of the pieces to see if they were rough fragments or smooth ones. There has been a wealth of recent studies suggesting that many plastic pieces are caused by the breakdown of larger pieces, and of course one of the problems of plastics is that they persist in the environment no matter their size—they just get smaller and thus more easily ingested.
4. My third question is what kind of plastic polymers or resins were found in this study (e.g., polyethylene, polypropylene, polyvinyl chloride, etc.). The authors describe their usef of the hot needle test to confirm that the pieces were indeed plastic. Different kinds of plastics have different physical and chemical properties (melting point, buoyancy, etc.) that could help researchers to determine which kinds of plastics are most common and most problematic. I am happy that the authors here categorized the microplastic pieces they found by shape (e.g., fibers, fragments, films) but they could have gone further.
5. My fourth question is whether they authors observed any assimilation or absorption of the tiny plastic pieces into the tissues of the cetaceans examined in this study. Even if they just looked at intestines, it would be possible to see if the fragments were just in the central intestinal lumen or if (upon gross or microscopic examination of the intestinal or other bodily tissues themselves) and microplastic fragments had passed through the intestinal wall by absorption, as has been documented in many marine species including some cetaceans. Did the authors consider this?
6. Although the writing is very good throughout the manuscript, the title seems strange to me. I would say cetaceans stranded on or along the Portuguese coast, not in the coast.
7. Table 1 (starting on Line 175) is useful, but it appears to need some editing. The right hand column does not list the full “Cause of death” heading—it says only “Cause of.” Also, the heading for the column about body length says just “Total,” and this is confusing. Why not use something like L for length or BL for body length (or TBL for total body length)? And why is the body length for the first specimen listed as -130? Shouldn’t this be 130? [Also, the units should be stated; I presume this is cm.]
The writing is good, although in some places the words are too bloated (for example, why not just say monitoring instead of "monitorization" in line 526?). There are other places where the language could be simplified: In the first sentence, this study characterizes (instead of "aims at characterizing") and similar instances throughout.
Round 2
Reviewer 1 Report
None